# Tumoricidal Activity and Side Effects of Radiolabeled Anti-NCAM [^131^I]-Iodine-ERIC1 in Neuroblastoma-Bearing Mice

**DOI:** 10.3390/ijms251910737

**Published:** 2024-10-05

**Authors:** Thomas Fischer, Felix Dietlein, Detlev Bongartz, Martin Klehr, Beate Zimmermanns, Matthias Schmidt, Angela Mohr, Fabian Mohr, Ferdinand Sudbrock, Philipp Krapf, Alexander Drzezga, Markus Dietlein, Klaus Schomäcker

**Affiliations:** 1Department of Nuclear Medicine, Faculty of Medicine and University Hospital Cologne, University of Cologne, Kerpener Str. 62, 50937 Cologne, Germany; thomas.fischer@uk-koeln.de (T.F.); beate.zimmermanns@uk-koeln.de (B.Z.); matthias.schmidt@uk-koeln.de (M.S.); ferdinand.sudbrock@uk-koeln.de (F.S.); p.krapf@fz-juelich.de (P.K.); alexander.drzezga@uk-koeln.de (A.D.); markus.dietlein@uk-koeln.de (M.D.); 2Computational Health Informatics Program, Boston Children’s Hospital, Harvard Medical School, Boston, MA 02115, USA; felix.dietlein@childrens.harvard.edu; 3Department of Pediatric Oncology and Hematology, Center for Integrated Oncology, University of Cologne, 50937 Cologne, Germany; detlev.bongartz@uk-koeln.de; 4Department of Pain Management, Helios Hospital, 47805 Krefeld, Germany; martin.klehr@helios-gesundheit.de; 5School of Mathematics and Science, Inorganic Chemistry, University Wuppertal, 42119 Wuppertal, Germany; amohr@uni-wuppertal.de (A.M.); fmohr@uni-wuppertal.de (F.M.); 6Forschungszentrum Jülich GmbH, Institute of Neuroscience and Medicine, Nuclear Chemistry (INM-5), Wilhelm-Johnen-Straße, 52428 Jülich, Germany; 7German Center for Neurodegenerative Diseases (DZNE), Bonn-Cologne, Venusberg-Campus 1/99, 53127 Bonn, Germany

**Keywords:** neuroblastoma, radioimmunotherapy, theranostics, targeted radiotherapy, antibodies

## Abstract

Preliminary studies on a radioactive antibody against the neural cell adhesion molecule (NCAM) demonstrated a significant accumulation of [^131^I]I-ERIC1 in neuroblastoma tumor cells in mice. This study aims to validate the therapeutic efficacy and potential adverse effects of these radioactive immunoconjugates (RICs) in neuroblastoma-bearing mice. To determine the highest tolerated dose, healthy SCID mice received 1 to 22 MBq of [^131^I]I-ERIC1, with the survival time measured. Tumor response was evaluated by administering 0.8 to 22 MBq of [^131^I]I-ERIC1 to neuroblastoma-bearing mice and assessing tumor size and systemic toxicity through body weight, blood counts, and survival. It was observed that doses up to approximately 3 MBq per animal (150 MBq/kg) were well tolerated, whereas higher doses resulted in systemic toxicity and death. The neuroblastomas exhibited a dose-dependent response, with optimal therapeutic efficacy achieved at 1.8–2.5 MBq per animal (90–125 MBq/kg), significantly extending survival by a factor of five. The antibody ERIC1 is a promising vehicle for the transport of beta emitters into NCAM-positive tumor tissue. An optimal dosage of the [^131^I]I-ERIC1 antibody can be established with a balance of tumor-static effects and adverse effects, resulting in a marked extension of survival time.

## 1. Introduction

Neuroblastoma (NB) is one of the most frequent solid extracranial tumors occurring in infancy. It occurs with a frequency of 1 in 7000 live births and an incidence of 10.54 cases per 1 million per year in children younger than 15 years with 90% of tumors diagnosed at an age < 5 years (median age at the time of diagnosis: 19 months) [1].

The main metastatic sites are regional lymph nodes, liver, bone, and bone marrow [2]. Despite multi-modal therapies such as chemotherapy, surgery, or radiation therapy, the prognosis of high-risk patients with NB remains poor, and the effective treatment of advanced NB remains a challenge for both clinicians and researchers [3,4,5] where, nowadays, theranostic appears to be a promising new option.

The term “theranostics” defines the development of more specific, individualized therapies that combine diagnostic and therapeutic capabilities into a single pharmaceutical agent. Radiotheranostics is a tailored application of theranostics in nuclear medicine using radionuclide-labeled substances (radiotheranostic agents) [6,7].

[^131^I]I-iodine, as a mixed gamma and beta emitter, has been a typical theranostic agent and the nuclide of choice in nuclear medicine therapy since the 1940s, particularly for both benign and malignant thyroid diseases. It emits beta particles, which provide a cytotoxic effect by damaging the DNA of targeted cancer cells, leading to cell death. Concurrently, I-131 emits gamma rays, which can be detected by imaging techniques such as single-photon emission computed tomography (SPECT) [8]. For neuroblastoma (NB), the classical radioactive vehicle for targeted radionuclide therapy (TRT) is, therefore, [^131^I]I-metaiodobenzylguanidine ([^131^I]I-MIBG). This treatment is well-established but is significantly limited by a low tumor response rate of 30–40% and considerable side effects (dose-limiting thrombocytopenia) [9,10], making alternative theranostic approaches with monoclonal antibodies (MABs or MoAbs) a reasonable consideration.

One of these promising MABs is ERIC1, a specific vector for the neural cell adhesion molecule (NCAM). NCAM is, when on normal lymphocytes, a characteristic marker for natural killer cells (NK cells) and in this context known under the synonym CD56 [11,12,13,14,15].

NCAM frequently occurs on the cell surface of many solid cancers [16,17,18,19,20] and is expressed on almost 100% of all NB cells [15,21,22,23]. The anti-NCAM MAB “ERIC1” accordingly reveals significant advantages over other MABs used against NB [12,14,15].

We had previously demonstrated that I-131-labeled ERIC 1 antibodies possess the ability to recognize NCAM-expressing tumor cells in vivo with high efficiency, suggesting their potential as promising agents for the diagnosis and treatment of NCAM-positive neuroblastoma tumors [24]. Motivated by these promising results, the tumor-inhibiting effect of [^131^I]I-ERIC1 was investigated in a preclinical feasibility study using a human neuroblastoma xenograft SCID mouse model. Radiation dose-dependent effects following the administration of [^131^I]I-ERIC1, such as animal weight and survival, were compared to non-exposed control animals.

## 2. Results

### 2.1. Labeling of the Antibody

The labeling of ERIC1 with I-131 achieved a yield of approximately 50–60%, and radiochemical purity was greater than 95%. In the case of [^131^I]I-ERIC1, the de-iodination rate was less than 5% over 6 h in saline solution. A specific activity of appr. 15 TBq/mmol was reached. Quality control by size-exclusion HPLC showed that the labeled antibody eluted after 6 to 7 min, while free [^131^I]I-iodide peaked at 10 to 11 min with radiochemical purity <95%.

### 2.2. Cohort 1: Dose Finding (Tumor-Free Mice)

#### 2.2.1. Group 1.1: Survival Time

The survival time was studied over a period of 21 days. The effects of the application of [^131^I]I-ERIC1 within the above-mentioned observation period showed that 100% of the tumor-free animals (N = 6) with injections of 16–22 MBq [^131^I]I-ERIC1 per mouse survived until day 6 with 100% and until day 8 with 50%, while 100% were dead after 8 days.

In the subgroup treated with 10–14 MBq per mouse (N = 6), the tumor-free animals survived until day 6 with 100% until day 7 with 66.6%, and until day 8 with 50%. On day 9, all animals were dead.

In the dose subgroup with 4–6 MBq per mouse (N = 6), the tumor-free animals survived until day 13 with 100%, until day 14 with 50%, and until day 17 with 17%.

All tumor-free animals in the subgroup treated with 1–2 MBq per mouse (N = 7) survived the complete observation period of 21 days.

#### 2.2.2. Group 1.2: Body Weight and Blood Count

No significant correlation was found between the activity dose applied and weight loss over 21 days in tumor-free and tumor-bearing animals of any of the treatment subgroups with the exception of the subgroup with 16–22 MBq/animal. In this case, the administration of the radiolabeled antibody led to a significant weight loss after 4 days p.i. by a factor of 1.2 compared to controls (92% of original weight) and after 9 days p.i. by a factor of 1.36 (74% of original weight).

Erythrocytes

The results are demonstrated in Figure 1.

No statistically significant changes were observed in the tumor-free subgroups administered 1–2 MBq or 4–6 MBq. At the higher radioactivity dose of 10–14 MBq, a decrease in erythrocyte levels to approximately 32% of the baseline value was observed after four days. Similarly, in the 16–22 MBq group, a non-statistically significant drop in erythrocyte levels to about 80% of baseline was noted by day 4. For both of these higher-dose groups, no observations were possible beyond the first week following RIC administration, as all animals had died by that time.

Leukocytes

The results are demonstrated in Figure 2.

No statistically significant results were observed regarding changes in leukocyte counts over time. In the group that received 1–2 MBq per animal, a decrease to 79% of the baseline value was suggested 14 days after RIC administration, followed by a partial recovery. A marked reduction to 39% of the baseline value was detected in the group administered 10–14 MBq per animal four days after RIC administration. This trend, however, could not be confirmed in the group with the highest radioactivity dose per animal.

Thrombocytes

The results are demonstrated in Figure 3.

Platelet counts in animals that received relatively low activity doses (1–2 MBq, 4–6 MBq) showed no significant time-dependent changes following RIC administration. However, in the groups with higher radioactivity per animal, a decrease to 39% of the baseline value (10–14 MBq per animal) or 68% of the baseline value (16–22 MBq per animal) was observed.

### 2.3. Cohort 2: Radioactivity Administration and Tumor Volume Measurement (Tumor-Bearing Mice)

#### 2.3.1. Group 2.1. Radiation-Induced Tumor Reduction by [^131^I]I-ERIC1 in Mice with Relatively Large Initial Tumor Diameter (46–173 mm^3^)

Growth factors for the different dose groups depending on the time after injection (day 1–day 8) are presented in Figure 4.

Eight days after injection, the growth factor in the LD1 groups was approximately 70% of the control group, while in the LD2 group, it was already 30%. In the HD group, we observed only a small additional effect compared to LD2. Only the animals treated with 1.8–2.5 MBq (LD2) per mouse survived the full 21-day observation time with a significantly reduced tumor growth.

As early as 5 days after RIC administration, evaluation by ANOVA (GraphPad Prism 10.3.0) revealed a significant difference between the HD group and the control group (*p* = 0.0464). By day 6 post-RIC, the LD2 group (*p* = 0.0112) also showed significant differences from the control group in addition to the HD group (*p* = 0.0015). Eight days after the administration of the RIC, all treatment groups exhibited significant differences in tumor growth behavior compared to the control group (Figure 4). Significant differences were also found within the treatment groups, specifically between the LD1 and LD2 groups, as well as between the LD1 and HD groups.

The evaluation of the period of duplication yielded a tumor doubling time of 2–3 d for the control groups. The doubling time was found to be 4 days for LD1 and 5–8 days for LD2. From the HD group, no tumor doubling time could be derived because, in all cases, a reduction in tumor masses was observed.

Starting from day 7 post-RIC administration, a significant correlation between the administered radioactivity per animal (control: 0 MBq, LD1: 1 MBq, LD2: 2 MBq, HD: 20 MBq) and tumor growth behavior (represented by the growth factor, GF) was observed (r = −0.3997, *p* (two-tailed): 0.0477). Similar results were observed on day 8 post-RIC administration (r = −0.4489, *p* (two-tailed): 0.0278).

#### 2.3.2. Group 2.2: Determination of Radiation-Induced Tumor Reduction by [^131^I]I-ERIC1 in Mice with Smaller Initial Tumor Diameter (10–40 mm^3^, Figure 5)

Both the control group (N = 5) and the LD2 group (N = 5) started with an initial tumor volume of 10–40 mm^3^. In the case of the control group, the termination criteria for tumor size were reached after 15 days, while in the case of the LD2 group, a reduction in the initial tumor size was observed. Additionally, after about 2 weeks, no tumor was any longer visible. After 42 days, the tumor growth started again, and, after 54 days, the termination criteria for tumor size were reached. It is noteworthy that only one mouse died after 23 days.

In the case of the control group, the termination criteria for tumor size were reached after 15 days, while in the LD2 group, a reduction in the initial tumor size was observed. Additionally, after approximately 2 weeks, no tumor was visible. After 42 days, tumor growth resumed, and, after 54 days, the termination criteria for tumor size were reached.

#### 2.3.3. Group 2.3. Radiation Effects of [^131^I]I-Iodide (Small Tumor Diameter)

After the administration of [^131^I]I-iodide in varying doses, no effect on tumor growth was observed. The treated animals behaved similarly to the control group.

#### 2.3.4. Group 2.4. Cold Antibody

The cold antibody without ^131^I showed no effect nor side effect in tumor-bearing mice (N = 5) when a dose of 1 mg per mouse was applied.

## 3. Discussion

NCAM presents a promising target for theranostics using monoclonal antibodies (MABs). In previous studies, we were able to demonstrate the first NCAM targeting in neuroblastoma-bearing mice using [^131^I]I-labeled ERIC1 [24]. The results of these studies highlight the significant potential of NCAM as a target structure for radioimmunotherapy.

The results presented here indicate a clear relationship between the applied radioactivity and the reduction in tumor growth, which is plausible. Statistically significant differences (*p* < 0.05) between the various treatment groups and the control group become apparent as early as five days post-administration of [^131^I]I-ERIC1.

Dose-determining experiments, combined with the measurement of dose-dependent parameters, such as survival time, animal weight reduction, and changes in blood count, allowed us to identify an optimal dose range of 1.8–2.5 MBq per mouse.

Considering an average initial mouse weight of 17 g, we can extrapolate a potential dose for juvenile patients. Applied to a patient weighing 15 kg, this would correspond to a dose of 1.6–2.2 GBq (105–147 MBq/kg), resulting in a lower administration compared to [^131^I]I-MIBG (444 MBq/kg) [25]. For comparison, studies on the radioimmunoconjugate [^131^I]I-UJ13A led to an optimal administered activity of 80 to 156 MBq/kg [26], while studies on [^131^I]I-3F8 reported an average administered activity of 144 MBq/kg [27]. The administration of different radioimmune compounds appears to have a similar activity range, as emphasized by the data presented here.

Of course, it would be presumptuous to derive a dosing recommendation for critically ill pediatric cancer patients based solely on this simple dose extrapolation. At this point, we would like to emphasize that significant modifications and further optimization of the RIC are necessary before clinical application can be considered. Additional studies, including thorough dose-escalation trials and comprehensive safety assessments, will be required to determine an appropriate and safe dosage regimen for human use. Therefore, it is premature at this stage to propose a suitable dose for clinical treatment without further extensive research and testing.

Further investigations showed that the initial tumor volume has a critical influence on the survival rate of mice. This conclusion aligns with the findings of Otto et al. [24] regarding the biokinetics of [^131^I]I-ERIC1, which demonstrated reduced specific accumulation in larger tumors. Extrapolating from the results with an initial tumor volume of 15 mm^3^ in an average 17 g mouse, this would correspond to an initial tumor burden of approximately 13 cm^3^ in a 15 kg patient. Applying the results of the optimal therapy dosage to the treatment of mice with an initial tumor volume of 15 mm^3^ led to a five-fold increase in survival time.

Additionally, the side effects described during the animal experiments were dose-dependent. Significant weight loss was observed only at the higher radioactivity of 20 MBq per mouse, corresponding to an activity of 18 GBq in humans. Severe side effects should, therefore, be expected at nearly ten times the activity suggested in this study (<2.2 GBq) and still three times higher than most MIBG protocols [28,29,30]. Greater sensitivity to radiation effects was observed in the blood-producing systems, consistent with the delayed RIC elimination from the blood reported by Otto et al. [24]. Regarding these side effects, our findings are limited to indicating a general trend rather than drawing definitive conclusions. However, this trend does support the dose-dependent nature of radiation-induced side effects. A decrease in blood cell counts was almost exclusively observed 4 h after RIC administration in animals that had received higher activities (10–14 MBq, 16–22 MBq). Unfortunately, further assessments at later time points were not possible as the animals had already succumbed. In the group that received 1–2 MBq, a tendency for a drop in erythrocyte and leukocyte counts was observed 14 days after RIC administration, followed by subsequent recovery.

Future experiments will explore how mice and tumor growth respond to fractionated multiple administrations of the radioimmunoconjugate. Due to the radiation-induced side effects of the administered RIC, only a small and well-adjusted dose range was acceptable. Fractionated application should, therefore, be considered to enhance the therapeutic outcome with fewer side effects.

In the group with the most promising results (1.8–2.5 MBq [^131^I]I-ERIC1 per mouse, initial tumor volume 15 mm^3^), the tumor recurred after 42 days. A repeated application of [^131^I]I-ERIC1 might overcome the risk of tumor recurrence without exposing patients to severe side effects.

## 4. Materials and Methods

### 4.1. Radiopharmaceutical

[^131^I]I-iodine as sodium iodide (7.4 GBq/mL; 370 MBq in 50 μL NaOH-solution) was obtained from Covidien GmbH, Neustadt, Germany.

### 4.2. Antibodies

ERIC1 antibodies were purified by protein G affinity chromatography from serum-free hybridoma supernatants (Hybridoma Express medium, PAA, Linz, Austria) as described elsewhere [15,24].

### 4.3. Labeling of ERIC1 with ^131^I

The antibody ERIC1 (0.1 mg/mL) was radiolabeled with varying activities of ^131^I using a variant of the chloramine-T method. A NAP-5 column (Amersham Biosciences; Piscataway, NJ; Sephadex G-25 medium) was used for purification. [^131^I]I-ERIC1 was obtained in radioactive concentrations of 10 to 200 MBq/mL.

Quality control of both I-131-labeled antibodies was carried out on 20 µL of labeling solution by size-exclusion HPLC (Column TosoHaas TSKgel 2000 SK; HPLC-system: Knauer, Berlin, Germany; radioactivity detector: model “Steffi”, Raytest, Straubenhardt, Germany, eluent: 0.09% NaCl solution). The radiochemical purity (RCR) obtained was >94%.

### 4.4. Animals

The animals used were 6-to-8-week-old female SCID mice (C.B-Igh-1b/IcrTac-Prkdcscid), supplied by Taconic M&B, Bomholt, Denmark, with no T- and B-lymphocytes but an intact NK (natural killer) cell system. All experiments were performed in accordance with national and institutional guidelines. All pain-inducing interventions, including tumor implantation, injection of radioactive substances, and killing of the animals, were performed under anesthesia with Ketanest S25 (PFIZER PHARMA PFE GmbH, Berlin, Germany).

### 4.5. Implantation of Xenografts

SCID mice were given intravenous injections of 20 µL (1 µg/µL) anti-asialo GM1 (anti-ASGM1) rabbit antiserum (WAKO Chemicals, Düsseldorf, Germany; dose recommended by the manufacturer) to deplete murine natural killer cells before the subcutaneous tumor-cell challenge. Mice were subcutaneously injected with 2 × 10^7^ human norepinephrine transporter (hNET)-expressing cells from the human NB cell line IMR5-75 to induce tumor xenograft growth. NCAM expression and on IMR5-75 cells was detected by flow cytometry as described elsewhere [24]. When tumors became macroscopically visible, the mice could be used for experiments.

### 4.6. Animal Experiments

A survey on the experimental design is provided in Table 1.

#### 4.6.1. Cohort 1: Dose Finding (Tumor-Free Mice)

The first cohort of tumor-free mice was divided into two groups consisting of four subgroups of 5–6 animals. Each group was treated with significantly different radioactivity levels of 1 to 2 MBq, 4 to 6 MBq, 10 to 14 MBq, and 16 to 22 MBq per animal. In total, 200 µL of a solution containing the respective amount of radioactivity as [^131^I]I-ERIC1 each containing 10 µg of non-radioactive ERIC1 was administered to the animals via tail vein. The second group contained an untreated control group of ten tumor-free mice for comparison.

Group 1.1: Survival time

The survival time of the experimental animals in the different dose subgroups was documented and evaluated. This allowed a dose range to be set in which the injected radioactivity is tolerated by the animals.

2.Group 1.2: Body weight and blood count

To determine the possible radiation-induced side effects of [^131^I]I-ERIC1, the body weight and blood count of the mice were compared to those of a control subgroup. The weight of the animals was determined daily on a digital balance (Sartorius AG PT 120 portable). The measurements took place two days after application to guarantee that no RIC was artificially removed from the body ahead of time by any blood sample. To do so, the tail tip of each test animal was resected. The resulting blood drop was collected with a heparin-coated capillary. Approximately 30–40 µL of blood was collected in an EDTA test tube. This was analyzed with COULTER^®^ Ac·T diff™ Analyzer (Beckman Coulter GmbH, Krefeld, Germany), and the number of erythrocytes, leukocytes, and thrombocytes in the blood count was determined. Each measurement was carried out in duplicate.

#### 4.6.2. Cohort 2: Radioactivity Administration and Tumor Volume Measurement (Tumor-Bearing Mice)

When the defined tumor volume was reached, 200 µL aliquots of [^131^I]I-ERIC1solution with the specified radioactivity was injected into the caudal vein of tumor-bearing SCID mice. The radioactivity of the whole mouse body was measured daily after the application of radioactivity using a radioisotope calibrator (ISOMED 1010 MED Nuklear-Medizintechnik Dresden GmbH, Dresden, Germany). One day before the application of [^131^I]I-ERIC1, the test animals were treated with 0.5 mg iodide to block the thyroid. This step was carried out with 0.5 mg “cold” iodide in 200 µL of water using a buttoned cannula inserted directly into the stomach. The tumor volume was measured with a micrometer screw gauge (PK-1025 Fa. Mitutoyo Japan, measurement precision 0.01 mm). After the immobilization of the mouse, the length, width, and thickness of the tumors were measured manually. To measure the thickness, the subcutaneous tumor was shifted to skin level. The tumor volume could thus be determined in mm^3^ assuming a spherical geometry.

Especially in the experiments with comparatively higher initial tumor volumes in subgroup 2.1, the variations in initial tumor volumes were notable, ranging from 46 to 173 mm^3^. To better compare the results of the RIC-mediated tumor reduction, a growth factor (GF) was established for each dose group. The GF was defined as the tumor volume at the time of examination V(t) divided by the initial tumor volume
V(0): GF(A) = (V(t))/(V(0))

Tumor reduction by [^131^I]I-ERIC1 (large tumor diameter)

Based on the results of the studies of the range of dose tolerated by the animals without any significant shortening of lifetime, Group 1 was divided into three dose subgroups and observed over a period of 21 days: two low-dose (LD) subgroups (0.8 MBq–1.5 MBq: LD1; 1.8 MBq–2.5 MBq: LD2), where radiation-induced death was not expected, as well as a third high-dose (HD) subgroup (16 MBq–22 MBq: HD) with a significant rate of radiation-induced death. The main interest in using this high radioactive dose (HD) was to study its tumoricidal effects during the animal’s lifespan, knowing well that radiation effects would reduce this.

2.Tumor reduction by [^131^I]I-ERIC1 (small tumor diameter)

In Group 2 (N = 5), the treatment was started at an earlier stage with a smaller initial tumor volume ranging between 10 mm^3^ and 40 mm^3^. The activity of [^131^I]I-ERIC1 selected for this experiment was based on that found in former experiments to achieve the most pronounced tumor reduction with the longest survival time (LD2: 1.8 MBq–2.5 MBq). In this way, the tumor-inhibiting effect of [^131^I]I-ERIC1 should be observed for the longest possible observation period.

The time dependence of tumor growth was also studied. Hereby, the period of the duplication of the tumor masses could be derived and, in the case of tumor shrinking, the half-life for the tumor reduction.

3.Radiation effects of [^131^I]I-iodide (small tumor diameter)

To ensure that the observed tumor-inhibitory effects were truly antibody-mediated and not due to nonspecific whole-body irradiation by ^131^I, the effects of non-tumor affine ^131^I-iodide were also examined. Four subgroups of test animals with smaller initial tumor diameter with 1–3 MBq (N = 5), 4–5 MBq (N = 5), and 9–13 MBq (N = 5) per animal were used. Application (200 µL per injection), thyroid blocking, and determination of tumor volume were carried out as described previously for [^131^I}I-ERIC.

4.Group 2.4: Comparison: Unlabeled antibody

To exclude interfering effects, i.e., therapeutic effects or possible side effects of the antibody, a group of five tumor-bearing animals was treated with 200 µL solution of ERIC1 with a significantly larger amount of antibody (1 mg ERIC1 per 1 mL) than used in the radioimmunoconjugates.

#### 4.6.3. Statistical Tests

For every dose subgroup (e.g., control, LD1, LD2, HD), the statistical tests ANOVA and Fisher‘s PLSD were performed. The growth factors (GFs) were determined on a daily basis over eight days. From day 0 to day 8, we thereby received the corresponding growth factors GF0–GF8 for all the four dose groups. It was statistically tested at which time point the GF (dependent variable) varied significantly depending on the dose (independent variable). A *p*-value less than 0.05 was considered statistically significant.

All data (N = 5–6) were analyzed, and values were presented as means ± SD. Statistical analyses were performed using GraphPad Prism 10.2.3 (Windows GraphPad Software, San Diego, CA, USA).

## 5. Conclusions

After the administration of the radioimmunoconjugate [^131^I]I-ERIC1 to neuroblastoma-bearing mice, a dose-dependent delay of tumor growth or even a tumor reduction was observed. With an initial tumor volume of 15 mm^3^ and a moderate radioactive dose of 1.8–2.5 MBq per mouse, the survival time of the treated animal was prolonged by a factor of five in comparison to controls. In addition, the radiation-induced side effects were dose-dependent. With radioactive doses greater than 4 MBq per mouse, the animals died of irreversible myelosuppression. Initial changes in blood count were reversible for doses of radioactivity of around 2.5 MBq per mouse. The antibody ERIC1 is, hence, a promising vehicle with tolerable side effects for the transport of a therapeutically effective beta emitter against NCAM-positive tumor tissue.

## Figures and Tables

**Figure 1 ijms-25-10737-f001:**
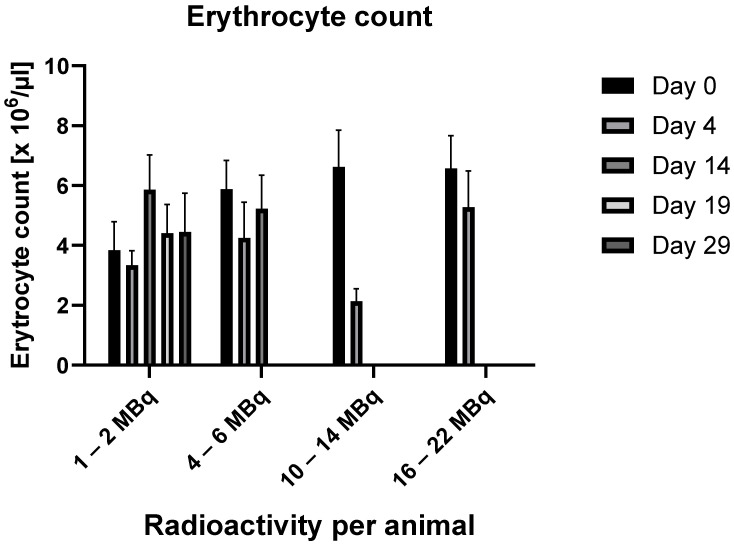
Changes in erythrocyte counts over time in relation to the administered radioactivity. (In the data group 4–6 MBq, values for days 19 and 29 are missing, and in the data groups 10–14 MBq and 16–22 MBq, values for days 14, 19, and 29 are missing because, by these time points, all animals had already died.)

**Figure 2 ijms-25-10737-f002:**
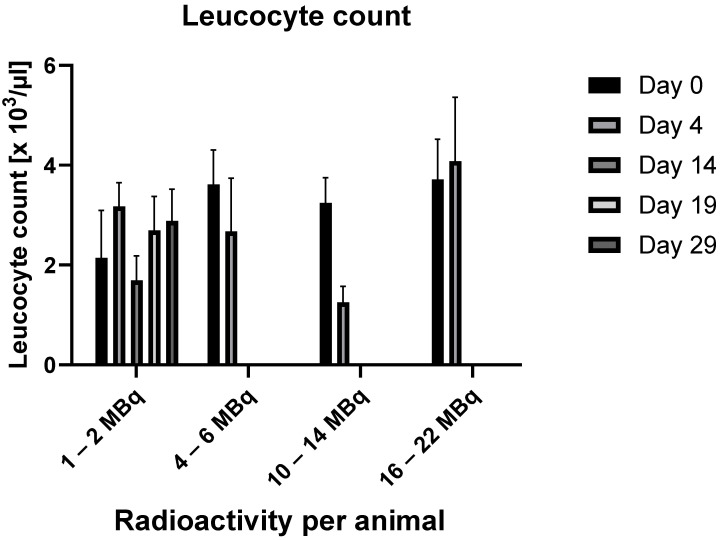
Changes in leucocyte counts over time in relation to the administered radioactivity. (In the data group 4–6 MBq, values for days 19 and 29 are missing, and in the data groups 10–14 MBq and 16–22 MBq, values for days 14, 19, and 29 are missing because, by these time points, all animals had already died.)

**Figure 3 ijms-25-10737-f003:**
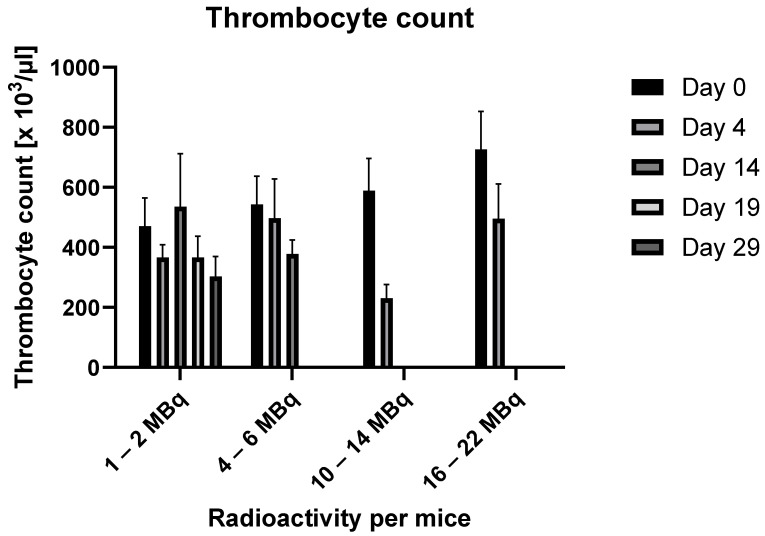
Changes in thrombocyte counts over time in relation to the administered radioactivity. (In the data group 4–6 MBq, values for days 19 and 29 are missing, and in the data groups 10–14 MBq and 16–22 MBq, values for days 14, 19, and 29 are missing because, by these time points, all animals had already died.)

**Figure 4 ijms-25-10737-f004:**
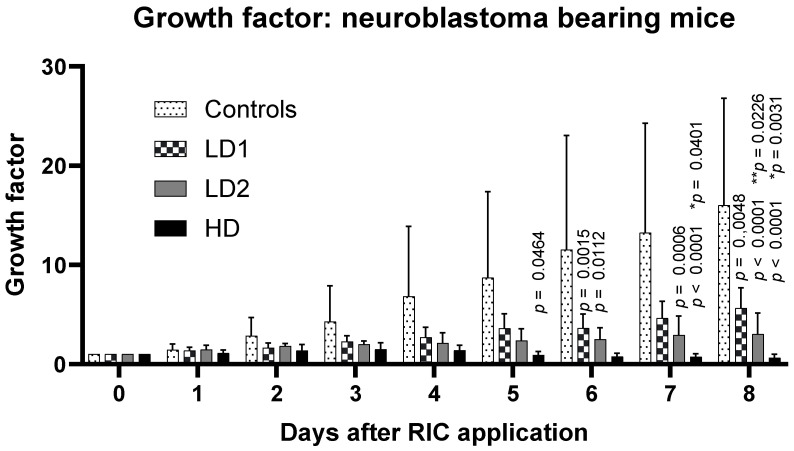
Dose-dependent effect on tumor response after the application of [^131^I]I-ERIC1. (*p*: comparison with controls; * *p*: HD vs. LD1, ** *p*: LD2 vs. LD1.) Abbreviations: LD1, low-dose group with 0.8–1.5 MBq [^131^I]I-ERIC1; LD2, low-dose group with 1.8–2.5 MBq [^131^I]I-ERIC1; HD, high-dose group with 16–22 MBq [^131^I]I-ERIC1; GF, growth factor.

**Figure 5 ijms-25-10737-f005:**
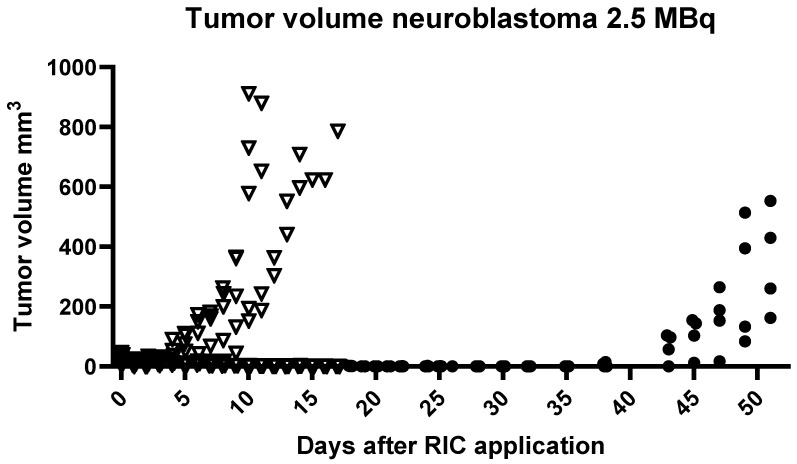
Examination of tumor volume over a prolonged observation period after administration of 1.8–2.5 MBq [^131^I]I-ERIC1 per mouse (initial tumor volume: 15–40 mm^3^) compared to a control group without RIC administration. Each point represents one animal (∇: controls, ∙: treated animals).

**Table 1 ijms-25-10737-t001:** Overview of the experimental design.

***Cohort 1*** ***Dose finding studies with tumor-free mice: Determination of dose dependent site effects***	**Groups/Objectives**	**Activities/Number**
	**1.1** **Determination of the survival time**	**1–2 MBq (N = 6)** **4–6 MBq (N = 6)** **10–14 MBq (N = 6)** **16–22 MBq (N = 6)** **No controls**
	**1.2** **Examination of the reduction in body weight and change in blood count**	**1–2 MBq (N = 5) in duplicate** **4–6 MBq (N = 5) in duplicate** **10– 14 MBq (N = 5) in duplicate** **16–22 MBq (N = 5) in duplicate** **Controls N = 10 ***
** *Cohort 2* ** *Animal experiments with **tumor-bearing mice:** Radioactivity administration and tumor volume measurement*	**Groups/Objectives**	**Activities/Number**
	2.1 Radiation-induced tumor reduction by [^131^I]I-ERIC1 in mice with relatively large initial tumor diameter (46–173 mm^3^)	0.8–1.5 MBq (N = 10) LD1 1.8–2.5 MBq (N = 4) LD2 16– 22 MBq (N = 5) HD Controls N = 20 **
	2.2 Determination of radiation-induced tumor reduction by [^131^I]I-ERIC1 in mice with smaller initial tumor diameter (10–40 mm^3^)	1.8–2.5 MBq (N = 5) Controls N = 5 **
	2.3 Investigations of the radiation effects of ^131^I-iodide in mice with smaller initial tumor diameter (10–40 mm^3^)	1–3 MBq (N = 5) 4–5 MBq (N = 5) 9–13 MBq (N = 5) Controls (N = 10) ***
	2.4 Investigation of the unlabeled antibody	N = 5

* Untreated tumor-free animals. ** Untreated tumor-bearing animals. *** Untreated tumor-bearing animals smaller initial tumor diameter.

## Data Availability

The original contributions presented in the study are included in the article, further inquiries can be directed to the corresponding author.

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
