# Peer review of "Tumoricidal Activity and Side Effects of Radiolabeled Anti-NCAM [131I]-Iodine-ERIC1 in Neuroblastoma-Bearing Mice"

_ijms, 2024, doi:10.3390/ijms251910737_

Round 1

Reviewer 1 Report

Comments and Suggestions for Authors

This paper is meaningful and may interest the researchers studying tumor therapies. Therefore, I would suggest to accept this paper after a minor revision.

  1. Labelling of the antibody - is there any quantitative data (graph) showing the measurement of labelling? The data should include error bar for statistical analysis.
  2. For Erythrocytes, leukocytes, thrombocytes - please also plot the data into a figure for comparison.
  3. Initial tumor volume - since this is mentioned by the author that it has a critical influence on the survival rate of mice, could the author also plot the data into a figure to demonstrate the correlation and how the extrapolating was calculated?

Author Response

[IJMS] Manuscript ID: ijms-3191699 - Major Revision

Tumoricidal activity and side-effects of radiolabeled anti-NCAM [131I]-Iodine-ERIC1 in neuroblastoma-bearing mice

Point to point reply

Reviewer 2

  • Line 81-83: The retention time of labeled antibodies should match the nonradiolabeled antibodies. Here there is a difference in time of 4 mins. Was the radiolabeled antibodies heavier than the nonradiolabeled antibodies?

We appreciate the reviewer's inquiry. First, I would like to bring attention to the exact wording of the passage in question: “Quality control by size exclusion HPLC showed that the labelled antibody eluted after 6 to 7 minutes while free [131I]I-iodide peaked at 10 to 11 minutes with radiochemical 82 purity < 95%..”

The discussion here focuses on the differences in retention times between free [131I]I-iodide and the labeled antibody, and not on the retention time differences between labeled and unlabeled antibodies. An additional point of consideration is why free radioactive iodide elutes after the radioiodinated antibody. This is due to the method employed, namely size exclusion HPLC.

In size exclusion chromatography (SEC), molecules are separated based on their size and shape. Larger molecules pass through the column more quickly because they cannot enter the pores of the stationary phase, while smaller molecules can penetrate the pores, causing them to elute later. In the case of radiolabeled iodide and radioiodinated antibodies, there is a distinct difference: Radiolabeled iodide (free iodide) is a small ion that can easily diffuse into the pores of the column's stationary phase. As a result, it has a longer retention time because it travels more slowly through the column, interacting with the pores.

Radioiodinated antibodies, on the other hand, are significantly larger molecules. Due to their size, they cannot enter the pores and therefore travel more quickly through the column, leading to a shorter retention time.

Thus, one can expect a clear difference in retention times: radioiodinated antibodies typically elute earlier, around 6-7 minutes, whereas free radiolabeled iodide elutes later, generally at 10-11 minutes. These differences in elution times are primarily due to the varying sizes of the molecules and their behavior within the stationary phase of the column.

  • The author must include a figure for the HPLC UV profile of [131I]I-ERIC1 immunoconjugate to show the peak at 6-7 min and [131I] peaked at 10-11 min.

We once again thank the reviewer for the valuable comment. In response, we would like to clarify that in the UV profile of the HPLC, the radioactive components cannot be distinguished. This differentiation is only possible in the radioactivity profile of the HPLC.

We can, however, include the radiochromatogram from the HPLC. It will appear in the manuscript under section 2.1, "Labeling of the antibody."

  • Line 79: The deiodination rate was observed only for 6 hours. Did the authors check the deiodination rate for multiple time points 12, 24, 48, 72, 84, 96, 108, and 120 hours? The author must perform an experiment to include the graph to show the deiodination rate at multiple points. Is it possible that during in vivo studies, the deiodination rate of less than 5% can be a major factor for low uptake and retention?

We appreciate the reviewer’s suggestion to extend the stability studies beyond 6 hours. However, we believe that such long-term stability assessments are not necessary from a radiobiological perspective. The key therapeutic effects of

The therapeutic effect of the I-131-labeled antibodies unfolds within the first few hours after the radioimmunoconjugate has localized to the tumor tissue. The main radiobiological mechanisms, including DNA damage such as single- and double-strand breaks, base modifications, and oxidative stress, are initiated shortly after the radioimmunoconjugate reaches the tumor site. Most of these critical cellular damages occur within the first 24 to 48 hours, triggering tumor cell death through apoptosis or necrosis. The therapeutic efficacy of the

[131I]I-labeled antibody is achieved rapidly because the emitted radiation primarily affects tumor cells during this early period. Therefore, the radioimmunoconjugate only needs to remain intact during the initial transport of the radioiodine to the tumor cells. Once the antibody has delivered the radioactive payload to the tumor, further degradation or deiodination does not significantly impact the treatment outcome.

Furthermore, with a deiodination rate of less than 5%, the retention of radioactivity in the tumor is sufficient to achieve the desired therapeutic effect. Extending the observation period beyond the early timeframe (12–120 hours) would not contribute meaningful insights, as the most significant radiobiological effects necessary for effective treatment occur within the first few hours after administration.

The radiobiological cascade following the initial damage is a critical process that determines the therapeutic effectiveness of the I-131-labeled radioimmunoconjugate. After the initial radiation-induced DNA damage - such as single- and double-strand breaks, base lesions, and oxidative damage - the tumor cells initiate a series of responses:

DNA Damage Response: This occurs within minutes to hours after the initial damage. The cell rapidly activates the DDR pathway, recruiting repair proteins like ATM, ATR, and DNA-PK to the site of damage. This phase typically lasts for up to 1-2 days, during which the cell attempts to repair the DNA lesions.

Cell Cycle Arrest: In response to extensive DNA damage, the cell may undergo a temporary or permanent arrest at key cell cycle checkpoints (G1/S or G2/M) to allow for repair. This arrest typically begins within hours of damage and can last for 1-3 days, depending on the extent of the injury and the cell’s ability to repair the damage.

DNA Repair or Apoptosis: If the damage is too severe, cells will either undergo repair or enter programmed cell death (apoptosis). The decision to repair or commit to apoptosis typically occurs within 1-3 days post-exposure. Apoptosis, marked by caspase activation and cell dismantling, usually completes within 24-48 hours after it is initiated.

Necrosis and Senescence: Cells unable to undergo apoptosis might experience necrosis, leading to uncontrolled cell death, often accompanied by inflammation. Necrosis can occur within 2-4 days after the damage. Alternatively, some cells enter a state of senescence, a permanent arrest in proliferation. Senescence can persist indefinitely, effectively stopping tumor growth for the lifespan of the affected cells.

Long-Term Effects: For cells that survive the initial damage and repair processes, genomic instability or other lingering effects may persist, leading to delayed cell death or dysfunction. These long-term effects can manifest over days to weeks following the initial exposure. Additionally, radiation-induced bystander effects, where neighboring cells experience damage due to signaling from irradiated cells, can extend the therapeutic impact over a period of several days to weeks.

  • What was the stability of [131I]I-ERIC1 immunoconjugate? The author should perform experiment to show the stability of [131I]I-ERIC1 immunoconjugate at different time point even after >120hr incubation.

This comment from the reviewer aligns with the point raised in item 3. We kindly ask that the radiobiological arguments provided in response to that point also be accepted in addressing this concern.

  • Instead of adding a table to show the significance of differences in tumor growth in neuroblastoma-bearing mice following treatment with varying doses of [131I]I-ERIC1 compared to an untreated control group (Table 1) the author must modify Figure 1 and include the p-value with significance in the graph.

We thank the reviewer for the valuable suggestion, and we will revise Figure 1 accordingly and remove Table 1. We have also revised the wording in the subsequent text accordingly.

  • Did the tumor start reoccurring after 35 days and was obvious at 42 days? The author should mention this clearly in their study. The author can't suggest that the dose of 1.8-2.5 Mbq/ mouse for [131I]I-ERIC1 could be used for treatment studies. The drug needs to be modified before it is administered to the patient.

We appreciate the reviewer’s important observation. We do not have experimental evidence to suggest that tumor regrowth had already begun at 35 days. We will revise the conflicting statement in line 233 of the discussion accordingly.

  • Line 179: The tumor regrowth is observed after 42 days. The circle represents the treated animals at 54 days and the tumor size has not declined at this point. The author must perform experiments at higher time points 90 days and 120 days to check the reduction in tumor size. Also, the use of 2.5 MBq [131I]I-ERIC1 per mouse can't be used as an optimal therapy for the treatment of tumors.

We appreciate the reviewer's comment regarding the need for experiments at extended time points, such as 90 and 120 days, to assess tumor reduction. However, we would like to clarify that our study adhered to predefined ethical endpoints for animal experimentation. Specifically, the study was designed to be terminated when one of the following criteria was met:

  • Tumor size reached approximately 530 mm³ (with an axis length of around 10 mm).
  • Any animal exhibited abnormal behavior, which could indicate significant distress or potential metastases, such as in the brain. These animals were humanely euthanized.
  • The animals succumbed to the disease.

In some cases, particularly within the control groups, the tumor grew rapidly between assessment intervals, occasionally surpassing the tumor size limit before the next scheduled examination. This accelerated tumor growth triggered the ethical endpoint, necessitating the termination of the experiment for those animals.

Given that these endpoints were established to minimize animal suffering and maintain ethical standards, continuing the experiments beyond 54 days would not have been feasible without violating these guidelines. We believe that the data collected up to this point are sufficient for evaluating the therapeutic efficacy of [131I]I-ERIC1 at the given dose. Further investigation into higher time points, though scientifically interesting, would require careful ethical consideration regarding the health and well-being of the animals.

“Also, the use of 2.5 MBq [131I]I-ERIC1 per mouse can't be used as an optimal therapy for the treatment of tumors.”

We sincerely thank the reviewer for this important observation. We fully agree that the dosage of 1.8-2.5 MBq/mouse for [131I]I-ERIC1, as used in our study, cannot be directly extrapolated for treatment in human patients, especially in critically ill children. While the preclinical results in animal models provide valuable insights into the potential efficacy and safety of this radioimmunoconjugate, they are by no means sufficient to establish treatment guidelines for pediatric patients.

We recognize that significant modifications and further optimization of the drug are required before any consideration for clinical application. Additional studies, including thorough dose-escalation trials and comprehensive safety assessments, will be essential to determine an appropriate and safe dosage regimen for human use. Therefore, we concur that it would be premature to suggest this dose as suitable for clinical treatment without further extensive research and testing.

We have therefore included the following passage in the discussion: “Of course, it would be presumptuous to derive a dosing recommendation for critically ill pediatric cancer patients based solely on this simple dose extrapolation. At this point, we would like to emphasize that significant modifications and further optimization of the RIC are necessary before clinical application can be considered. Additional studies, including thorough dose-escalation trials and comprehensive safety assessments, will be required to determine an appropriate and safe dosage regimen for human use. Therefore, it is premature at this stage to propose a suitable dose for clinical treatment without further extensive research and testing.”

  • At several places the author has not included any table/figure to represent the results. The author should include the result of the radiation effect of [131I]-Iodine on no change in tumor diameter (image of tumor) or figure for cold antibody study?

We sincerely appreciate the reviewer’s thoughtful suggestion. However, we would prefer to avoid including additional figures or tables for the [131I]I-Iodine and cold antibody studies, as they do not provide any further information beyond what is already described in the text. Both interventions showed no impact on tumor growth, and their results were identical to those of the control group. As such, we believe that the brief verbal description sufficiently conveys the findings without the need for additional visual representation.

That being said, we are committed to clarity and transparency in our reporting, and if the reviewer feels that such visuals are essential, we are open to further discussion.

Reviewer 2 Report

Comments and Suggestions for Authors

The treatment of neuroblastoma by using NCAM as a characteristic marker for natural killer cells. The author of the current study has developed [131I]I-ERIC1 to diagnose and treat NCAM-positive cells.

The introduction, material and method, results and discussion are well written. 

However, the author should include the following experiments to improve the study:

Major comments:

1) Line 81-83: The retention time of labeled antibodies should match the nonradiolabeled antibodies. Here there is a difference in time of 4 mins. Was the radiolabeled antibodies heavier than the nonradiolabeled antibodies?

2) The author must include a figure for the HPLC UV profile of [131I]I-ERIC1 immunoconjugate to show the peak at 6-7 min and [131I] peaked at 10-11 min.

3) Line 79: The deiodination rate was observed only for 6 hours. Did the authors check the deiodination rate for multiple time points 12, 24, 48, 72, 84, 96, 108, and 120 hours? The author must perform an experiment to include the graph to show the deiodination rate at multiple points. Is it possible that during in vivo studies, the deiodination rate of less than 5% can be a major factor for low uptake and retention?

4) What was the stability of [131I]I-ERIC1 immunoconjugate? The author should perform experiment to show the stability of [131I]I-ERIC1 immunoconjugate at different time point even after >120hr incubation.

5) Instead of adding a table to show the significance of differences in tumor growth in neuroblastoma-bearing mice following treatment with varying doses of [131I]I-ERIC1 compared to an untreated control group (Table 1) the author must modify Figure 1 and include the p-value with significance in the graph.

6) Did the tumor start reoccurring after 35 days and was obvious at 42 days? The author should mention this clearly in their study. The author can't suggest that the dose of 1.8-2.5 Mbq/ mouse for [131I]I-ERIC1 could be used for treatment studies. The drug needs to be modified before it is administered to the patient.

7) Line 179: The tumor regrowth is observed after 42 days. The circle represents the treated animals at 54 days and the tumor size has not declined at this point. The author must perform experiments at higher time points 90 days and 120 days to check the reduction in tumor size. Also, the use of 2.5 MBq [131I]I-ERIC1 per mouse can't be used as an optimal therapy for the treatment of tumors.

8) At several places the author has not included any table/figure to represent the results. The author should include the result of the radiation effect of [131I]-Iodine on no change in tumor diameter (image of tumor) or figure for cold antibody study?

9) Discussion: 

While discussing the author must mention that the optimal dose of 1.8-2.8 MBq [131I]I-ERIC1 per mouse is suitable for an initial tumor volume of 10 – 40 mm³ and not for larger volumes. So if the author is suggesting a dose of 1.6 – 2.2  GBq (105 – 147 MBq/kg) it should be in children and when the tumor is small. Even I think that the author needs to modify the immunoconjugate and perform more experiments before concluding that the immunoconjugate can be administered at the mentioned dose in patients. There are side effects of the immunoconjugate.

10) Material and Method: The material and method is written very clearly.

Animals: Line 255

Why the author has only included females in their study? What will be the effect on males? The author must include in the study checking the effect of [131I]I-ERIC1 on both males and females. Then the study will not be sex-biased.

Minor comment:

1) Spelling mistake: Line 97: weight

2) What is p.i.? The author should include the full form before mentioning the short form.

3) Line 137-138 can be added beneath the Figure1 heading.

4) Figure 2 heading 2.5 MBq/mouse instead of 2,5.

5) The author should mention the correct tumor size at all places, In line 165 the initial tumor size is 10-40 mm³ whereas in line 174 the initial tumor volume: 15 – 40 mm³. This is confusing to the readers.

6)In Line 233 the author has mentioned the tumor recurred after 35 days, it should be similar in the complete study.

Author Response

(The authors gave the same response as above.)

Round 2

Reviewer 2 Report

Comments and Suggestions for Authors

The authors could improve Fig1, 2, 3. the spacing between the bars are not appropriate. the spacing should be equal for each bar represnted.

Author Response

[IJMS] Manuscript ID: ijms-3191699 - Major Revision

Tumoricidal activity and side-effects of radiolabeled anti-NCAM [131I]-Iodine-ERIC1 in neuroblastoma-bearing mice

Point to point reply

Reviewer 2

The authors could improve Fig1, 2, 3. the spacing between the bars are not appropriate. the spacing should be equal for each bar represented.

Thank you very much for this important suggestion. Unfortunately, it is not straightforward to adjust the positioning of individual bars in figures created with GraphPad Prism. However, we have made efforts to clarify the spacing between the different radioactivity groups. In the groups starting from 4–6 MBq, values for the later time points are missing because all animals had already died by then. The following explanation has been added in parentheses to the legends of Figures 1–3: (In the data group 4–6 MBq, values for days 19 and 29 are missing, and in the data groups 10–14 MBq and 16–22 MBq, values for days 14, 19, and 29 are missing because, by these time points, all animals had already died). We hope that the adjustments we have made improve the overall clarity of the figures.
